# Prognostic Significance of Modified Shine and Lal Index in Patients with Non-Small Cell Lung Cancer Undergoing Surgical Resection

**DOI:** 10.3390/biomedicines13040937

**Published:** 2025-04-11

**Authors:** Soomin An, Wankyu Eo, Sookyung Lee

**Affiliations:** 1Department of Nursing, Dongyang University, Yeongju 36040, Republic of Korea; 2College of Medicine, Kyung Hee University, Seoul 05278, Republic of Korea; 3Department of Clinical Oncology, College of Korean Medicine, Kyung Hee University, Seoul 05278, Republic of Korea

**Keywords:** carcinoma, non-small cell lung cancer, mean corpuscular volume

## Abstract

**Background**: Although white blood cell-related indices are established prognostic markers in lung cancer, the prognostic significance of red blood cell (RBC) indices—mean corpuscular volume (MCV), mean corpuscular hemoglobin (MCH), and mean corpuscular hemoglobin concentration (MCHC)—remains unclear. This study assessed the prognostic value of RBC indices for predicting survival outcomes in patients who underwent curative-intent surgery for stage I–IIIA non-small cell lung cancer (NSCLC). **Methods**: This retrospective analysis of 437 patients evaluated the prognostic significance of MCV, MCH, MCHC, and the modified Shine and Lal Index (mSLI), calculated as (MCV^2^ × MCH) × 0.0001, using Cox regression analysis. Model performance was evaluated using various metrics, including the concordance index (C-index) and integrated discrimination improvement (IDI). **Results**: In the multivariate Cox regression analysis, each RBC index was tested separately as an overall survival (OS) predictor in models that consistently included age, American Society of Anesthesiologists Physical Status (ASA-PS), pleural invasion, tumor–node–metastasis (TNM) stage, and the Noble and Underwood (NUn) score. Given its superior predictive performance, the mSLI model, which incorporates mSLI in addition to other covariates, was finalized and outperformed the baseline TNM staging model (C-index: 0.840 vs. 0.708, *p* < 0.001) and demonstrated significant improvements in IDI at 3 and 5 years (*p* < 0.001). Compared to the intermediate model—comprising the same covariates as the mSLI model except for mSLI—the mSLI model showed a slightly higher C-index (0.840 vs. 0.835, *p* = 0.554) and significant improvements in IDI at 3 years (*p* = 0.008) and 5 years (*p* = 0.020). **Conclusions**: mSLI was an independent prognostic marker for OS in stage I–IIIA NSCLC, enhancing risk stratification and providing incremental predictive value beyond that of traditional models. Incorporating mSLI into prognostic frameworks may improve clinical decision-making. However, external validation is required to confirm its clinical utility.

## 1. Introduction

Surgical resection remains the standard of care for patients with stage I–IIIA non-small cell lung cancer (NSCLC) [1,2]. However, postoperative pulmonary complications and suboptimal survival outcomes persist as critical challenges, emphasizing the need for improved prognostic markers to identify high-risk individuals and support personalized treatment strategies [1,2].

Although the tumor–node–metastasis (TNM) staging system is the primary tool for assessing the prognosis of NSCLC [3,4,5,6], it does not fully account for disease heterogeneity. Many patients with the same TNM stage exhibit widely varying survival outcomes, highlighting the necessity for additional biomarkers to refine prognostic models and enhance clinical decision-making [7,8].

Demographic factors such as age, sex, performance status, American Society of Anesthesiologists Physical Status (ASA-PS), and smoking history are established predictors of survival in NSCLC [3,5,9,10]; however, they have inherent limitations. Similarly, clinicopathological features, including tumor histology, size, pleural/vascular/lymphatic/perineural invasion, and residual disease, offer valuable insights into survival predictions [2,3,4,5,6,10,11,12,13]. However, they do not capture systemic processes such as inflammation or immune responses that can affect prognosis.

Biochemical markers have recently gained attention as potential prognostic tools in NSCLC. Markers such as C-reactive protein (CRP) and serum albumin (ALB) and combined indices such as the CRP-to-ALB ratio (CAR) have demonstrated strong associations with survival outcomes [14,15,16]. Similarly, white blood cell (WBC)-derived markers, including the neutrophil-to-lymphocyte ratio (NLR) and lymphocyte-to-monocyte ratio (LMR), provide additional insights into the systemic inflammatory responses that drive cancer progression [17,18]. Combined indices that integrate biochemical and WBC-derived markers, such as the prognostic nutritional index (PNI), CRP-to-lymphocyte ratio (CLR), CRP–ALB–lymphocyte (CALLY) index, and Noble and Underwood (NUn) score, have also demonstrated a predictive value for overall survival (OS) in NSCLC [9,19,20,21]. However, these indices require further validation in larger and more diverse populations.

In contrast to WBC- and platelet-derived markers, red blood cell (RBC)-derived markers remain relatively underexplored in the prognostic evaluation of NSCLC. Malignancies, including NSCLC, are frequently accompanied by anemia and chronic inflammation [22,23,24], both of which can significantly impact RBC morphology and function. These alterations in RBC parameters may not only reflect the presence of cancer-related anemia but also serve as indirect indicators of tumor burden and systemic disease activity. As NSCLC is one of the most common and aggressive cancers worldwide, its influence on RBC physiology may hold prognostic value. Evaluating RBC-derived markers could provide novel insights into disease progression and patient outcomes, potentially enhancing current prognostic models.

Several studies have linked red cell distribution width (RDW) to survival outcomes in NSCLC [25,26,27,28,29,30]. Additionally, the hemoglobin-to-RDW ratio has been identified as a significant prognostic factor in NSCLC [30,31]. These findings suggest that RBC-derived markers directly influence tumor biology or serve as surrogate indicators of systemic physiological changes associated with NSCLC.

Among RBC indices, mean corpuscular volume (MCV) and mean corpuscular hemoglobin concentration (MCHC) have been identified as independent prognostic indicators of OS in NSCLC [32,33]. However, clinical studies focusing on RBC indices are scarce, and the prognostic value of mean corpuscular hemoglobin (MCH) is unclear. Moreover, the potential role of combined RBC indices that may offer additional prognostic value has not been thoroughly investigated.

The Shine and Lal Index (SLI), originally developed as a diagnostic tool for thalassemia, is a composite measure derived from MCV and MCH [34,35,36]. While its role in NSCLC prognosis has not been established, its nature as a combined RBC index suggests potential utility in survival prediction.

This study aimed to evaluate the prognostic significance of RBC indices, including MCV, MCH, and MCHC, and a combined RBC index, the Shine and Lal Index (SLI), in patients who underwent curative-intent surgical resection for stage I–IIIA NSCLC.

In addition, we sought to determine whether incorporating these indices into a traditional prognostic model could improve OS prediction, enhance risk stratification, and guide personalized treatment strategies.

## 2. Materials and Methods

### 2.1. Patients

This retrospective study analyzed electronic medical records of patients with NSCLC who underwent surgical resection at Kyung Hee University Hospital in Gangdong between January 2010 and February 2024. Standard staging procedures included chest and abdominopelvic computed tomography (CT) and positron emission tomography-CT.

Eligibility criteria included a confirmed diagnosis of primary NSCLC [37], stage I–IIIA disease [38], and complete resection with negative microscopic margins (R0 resection) [39]. Exclusion criteria encompassed prior anticancer therapy for NSCLC, advanced disease (stage IIIB–IV), presence of concurrent secondary malignancies or a history of malignancy within the past 5 years, and active infections or autoimmune/connective tissue diseases requiring treatment.

After surgical resection, the patients with stage II or IIIA NSCLC received adjuvant chemotherapy with a platinum-based doublet regimen. Surveillance after treatment consisted of contrast-enhanced chest and abdominopelvic CT scans every 3–6 months during the first three years, followed by chest CT alone thereafter.

This study was approved by the Institutional Review Board of Kyung Hee University Hospital at Gangdong (no. 2024-12-015), which waived the requirement for informed consent owing to the retrospective nature of the study.

### 2.2. Clinical Characteristics

The clinicopathological variables analyzed in this study included age, sex, smoking history, alcohol consumption, body mass index (BMI), ASA-PS, type of surgery, histology, tumor size, invasion status (pleural/vascular/lymphatic/perineural), TNM stage, and microscopic residual disease status. Alcohol consumption was defined as drinking more than once per week, regardless of quantity [40]. Pleural invasion was assessed and assigned a score between 0 and 3, reflecting increasing severity [41].

Laboratory tests assessed in this study comprised standard blood chemistry measures, such as CRP and liver function tests (total protein, ALB, total bilirubin [TB], aspartate aminotransferase [AST], and alanine aminotransferase [ALT]). Hematological parameters assessed in this study included total WBC count, absolute neutrophil count (ANC), absolute monocyte count (AMC), absolute lymphocyte count (ALC), RBC count, hemoglobin level, MCV, MCH, MCHC, and platelet count. Laboratory results were based on blood samples obtained no more than seven days before surgery [42,43].

We also assessed the prognostic values of various combinations of these indices. LMR, NLR, and platelet-to-lymphocyte ratio (PLR) were calculated as follows: ALC/AMC, ANC/ALC, and platelet count/ALC, respectively. Additionally, the CAR, PNI, CLR, CALLY index, and the NUn score were calculated as previously reported [9,16,20,21,44]. The SLI was originally calculated using the formula SLI = MCV^2^ × MCH × 0.01 [34,35]. However, this calculation often yields values in the thousands or higher range, which can obscure prognostic significance, especially when analyzed as a continuous variable in prognostic studies. To improve usability, the SLI was modified by multiplying an additional factor of 0.01, resulting in the modified Shine and Lal Index (mSLI), calculated as (MCV^2^ × MCH) × 0.0001. This adjustment reduced the numerical range while retaining clinical relevance, enabling clearer interpretation and application in clinical and research settings.

### 2.3. Statistical Analyses

OS was defined as the time from surgical resection to all-cause mortality. Continuous variables were analyzed without categorization to maintain data richness and enhance generalizability [45,46,47].

Associations between clinical variable pairs were assessed using Spearman’s correlation analysis. The results were visualized using a chord diagram, highlighting significant associations between variable pairs with correlation coefficients >0.5 or <−0.5. Generalized Additive Models (GAMs) with the gamma family were employed to model the relationship between the dependent variable and predictors (independent variables) identified through Least Absolute Shrinkage and Selection Operator (Lasso) regression.

Cox proportional hazards regression models were used to identify independent predictors of OS and to estimate the corresponding hazard ratios (HRs). This method was chosen for its ability to simultaneously adjust for multiple covariates and to provide interpretable estimates of the relative risk associated with each prognostic factor.

Covariates with a *p*-value less than 0.20 in the univariate analysis were considered for entry into the multivariate Cox regression model, with the exclusion of variables that did not satisfy the proportional hazards assumption. Multicollinearity was assessed using variance inflation factor (VIF) values to confirm model stability. Relationships between significant continuous variables and log-relative hazards were further explored using fractional polynomial curves. This method allowed for visual exploration of their relationship with mortality risk, which may not be readily apparent in tabular data.

Model performance was evaluated based on the concordance index (C-index), and differences between models were statistically tested using 1000 bootstrap resamples. Time-dependent areas under the curves (AUCs) at 3 and 5 years postoperatively were calculated to compare the predictive capabilities of the models, with robustness ensured through 1000 bootstrap resamples. Model comparisons were further conducted using integrated discrimination improvement (IDI) to quantify gains in risk discrimination and decision curve analysis (DCA) to evaluate the clinical net benefit of each model across a range of threshold probabilities.

Time-dependent C-index curves were also generated to evaluate and compare the predictive performance of each model throughout the follow-up period.

A nomogram was developed to predict OS using the final model, and its accuracy was validated using calibration curves generated from 1000 bootstrap samples. All analyses were performed using R (version 4.1.2) with statistical significance set at *p* < 0.05.

Propensity score matching was not applied, as the study aimed to identify prognostic factors and develop predictive models rather than compare treatment groups. This objective was appropriately addressed using multivariate Cox regression, which adjusts for key covariates and minimizes confounding.

## 3. Results

### 3.1. Clinicopathological Characteristics of the Patients

In the present study, a total of 36 variables were included, comprising both previously reported prognostic factors and additional variables selected for exploratory analysis to ensure comprehensive and consistent results. A total of 437 patients were evaluated, of whom 428 (97.9%) were Asian and 9 (2.1%) were Caucasian. The sample size was determined based on strict inclusion and exclusion criteria, as outlined in the Section 2, to achieve a homogeneous study population with adequate statistical power for assessing survival outcomes. To minimize potential biases related to missing data, only patients with complete clinical and laboratory records were included in the final analysis.

Most patients underwent lobectomy or its variants (65.9%, *n* = 288), followed by segmentectomy (32.7%, *n* = 143) and pneumonectomy (1.4%, *n* = 6). Adenocarcinoma was the most common histological subtype, diagnosed in 320 patients (73.2%), followed by squamous cell carcinoma in 95 patients (21.7%). The less common subtypes were pleomorphic (2.3%, *n* = 10), adenosquamous (1.1%, *n* = 5), large cell (0.7%, *n* = 3), and large cell neuroendocrine (0.9%, *n* = 4) carcinomas. According to the disease stage, 68.6% (*n* = 300) had stage IA or IB, 18.1% (*n* = 79) had stage IIA or IIB, and 13.3% (*n* = 58) had stage IIIA (Table 1).

### 3.2. Associations of mSLI with Clinical and Laboratory Parameters

Spearman’s correlation analysis revealed strong positive correlations between mSLI and MCV (*r* = 0.98) and MCH (*r* = 0.93). In contrast, no significant correlations (defined as |*r*| > 0.5) were observed between mSLI and other clinical or laboratory variables, including age, BMI, WBC, ANC, AMC, ALC, RBC count, hemoglobin concentration, MCHC, platelet count, LMR, NLR, PLR, liver function tests (total protein, ALB, AST, ALT, and TB), CRP, CAR, PNI, CLR, CALLY index, or NUn score (Figure 1).

To identify predictors of mSLI, Lasso regression was applied to evaluate potential predictors of mSLI, including categorical variables such as sex, smoking history, alcohol consumption, ASA-PS, histology, TNM stage, and pleural/vascular/lymphatic/perineural invasion, as well as continuous variables mentioned above. From this analysis, eight variables—sex, alcohol consumption, TNM stage, lymphatic invasion, ASA-PS, hemoglobin concentration, MCV, and MCH—were identified as significant predictors of mSLI. A GAM with a gamma family further refined the selection, retaining only MCV and MCH as significant predictors in the final model. The intercept of the GAM was 3.253 (*p* < 0.001), confirming statistical significance. For the smooth terms, the estimated degrees of freedom for s(MCV) (8.995) and s(MCH) (8.997) remained within their predefined basis dimensions (reference degrees of freedom = 9), ensuring sufficient model flexibility. Both predictors demonstrated highly significant associations with mSLI (*p* < 0.001), with their smooth terms exhibiting a consistent upward trend. The model demonstrated excellent performance, achieving an adjusted R^2^ of 1 and explaining 100% of the deviance, underscoring its accuracy in capturing the relationship between the predictors and mSLI (Figure 2).

In conclusion, mSLI was strongly associated with MCV and MCH, whereas no significant associations were observed with demographic, pathological, or other clinical variables.

### 3.3. Cox Proportional Hazard Regression Analysis

Patients were followed for a median duration of 45.8 months (interquartile range: 22.7–75.7 months). Univariate Cox regression identified several variables significantly associated with OS, including age, sex, smoking history, ASA-PS, histological subtype, tumor size, pleural/vascular/lymphatic/perineural invasion, TNM stage, WBC count, LMR, NLR, ALB, CRP, CAR, PNI, CLR, CALLY index, and NUn score. These findings reinforce the prognostic relevance of traditional staging and histology while also highlighting the added value of inflammatory and nutritional biomarkers. All identified variables were subsequently entered into the multivariate Cox regression analysis (Table 2).

Among the RBC indices, the mSLI, MCV, and MCH met the inclusion criterion and were incorporated into the multivariate model. Conversely, MCHC was excluded due to its lack of statistical significance (Table 2).

The prognostic utility of RBC indices (mSLI, MCV, and MCH) was assessed by sequentially incorporating them into multivariate models. When mSLI was included without MCV and MCH, it remained a significant predictor (HR = 1.08, *p* = 0.003). In addition to mSLI, age (HR = 1.07, *p* < 0.001), ASA-PS (HR = 1.92, *p* = 0.009), pleural invasion (HR = 1.45, *p* = 0.002), TNM stage (HR = 3.37, *p* < 0.001), and the NUn score (HR = 1.96, *p* < 0.001) were significant covariates, forming the mSLI model. This model demonstrated a strong predictive performance, with a C-index of 0.840. The VIF values indicated minimal collinearity among the final predictors: age (1.03), ASA-PS (1.06), pleural invasion (1.10), TNM stage (1.11), mSLI (1.10), and NUn score (1.14) (Table 3).

When MCV was included without MCH and mSLI, it remained a significant predictor (HR = 1.06, *p* = 0.012). The MCV model, which incorporated MCV with age, ASA-PS score, pleural invasion, TNM stage, and NUn score, demonstrated a C-index of 0.837. Similarly, when MCH was included without MCV and mSLI, it also remained significant (HR = 1.18, *p* = 0.007). The MCH model incorporating MCH along with the same five covariates exhibited a C-index of 0.837 (Table 3).

Although the C-index of the mSLI model was slightly higher than that of the MCV model (*p* = 0.430) and the MCH model (*p* = 0.828), these differences were not statistically significant. These findings suggest that while MCV and MCH independently contribute to prognosis, mSLI may offer a slightly greater prognostic value for OS in patients with NSCLC.

We generated time-dependent C-index curves to evaluate and compare the predictive performance of the three models (mSLI, MCV, and MCH) across the follow-up period. In the early phase (12–24 months), the mSLI model demonstrated performance nearly identical to the MCH model, with all three models achieving high C-index values. During the mid-term period (36–48 months), the mSLI model showed marginal superiority, either matching or slightly surpassing the MCV and MCH models. At 60 months, although all models showed a decline in performance, the mSLI model maintained a slight advantage (0.827 vs. 0.825 for MCV and 0.824 for MCH), indicating better long-term stability. Overall, the mSLI model maintained consistently high C-index values throughout the 5-year follow-up, often ranking first or tying for the best performance at each time point. These small but consistent advantages suggest that the mSLI model offers greater robustness and reliability for long-term survival prediction in patients with NSCLC (Figure 3).

When all the RBC indices (mSLI, MCV, and MCH) were included in the multivariate analysis, both MCV and mSLI remained significant. However, substantial collinearity was observed between these variables (VIF: 36 for mSLI and 35 for MCV), indicating redundancy. Given its higher C-index and superior model performance, the mSLI model was finalized.

Fractional polynomial analysis of the mSLI model revealed linear relationships between the log relative hazard and the continuous variables (age, mSLI, and NUn score). This finding suggests that each of these factors has a direct and proportional impact on survival outcomes, with the hazard increasing predictably as their values rise (Figure 4).

### 3.4. Comparison Between mSLI and Baseline Models

The mSLI model exhibited significantly improved predictive performance compared with the baseline model, which relied solely on TNM staging. The C-index of the mSLI model was significantly higher than that of the baseline model (0.840 vs. 0.708; *p* < 0.001) (Table 4).

In terms of time-dependent performance, the mSLI model achieved significantly higher AUC values for predicting OS at both 3 years (0.878 vs. 0.734; *p* < 0.001) and 5 years (0.850 vs. 0.708; *p* < 0.001), reflecting enhanced discriminatory capability. Additionally, the IDI further supported the superiority of the mSLI model, with substantial gains at both 3 years (IDI = 0.241, *p* < 0.001) and 5 years (IDI = 0.218, *p* < 0.001) (Table 4).

DCA reinforced these findings, showing that the mSLI model consistently provided a greater net clinical benefit across a range of high-risk thresholds for predicting 3- and 5-year OS (Figure 5). A higher net benefit signifies that the model more effectively identifies high-risk patients who are likely to experience adverse outcomes, thereby facilitating timely and targeted interventions, ultimately leading to improved patient outcomes. These results underscore the potential clinical utility of the mSLI model in optimizing individualized patient management and improving survival outcomes.

### 3.5. Comparison Between mSLI and Intermediate Models

The mSLI model demonstrated slightly superior predictive performance compared with the intermediate model, which excluded mSLI while retaining all other variables from the mSLI model. Although the C-index of the mSLI model was marginally higher than that of the intermediate model (0.840 vs. 0.835), the difference was not statistically significant (*p* = 0.554). At 3 years, the AUC of the mSLI model for OS was slightly greater than that of the intermediate model (0.878 vs. 0.873, *p* = 0.614), with a similar trend observed at 5 years (0.850 vs. 0.846, *p* = 0.662); however, these differences were not statistically significant (Table 4).

In contrast, the IDI metric revealed that the mSLI model offered significantly enhanced discriminatory ability over the intermediate model, with higher IDI values observed at both 3 years (IDI = 0.047; *p* = 0.008) and 5 years (IDI = 0.029; *p* = 0.020) (Table 4). These findings highlight the added prognostic value of incorporating mSLI into the multivariable model, suggesting that it contributes meaningfully to risk stratification beyond traditional and laboratory parameters (Table 4).

### 3.6. Nomogram for Predicting 3- and 5-Year Survival

A nomogram was developed from the mSLI model, incorporating mSLI and additional clinical variables to estimate 3- and 5-year survival outcomes (Figure 6). The calibration plots demonstrated strong agreement between the predicted and observed survival outcomes, supporting the accuracy and clinical applicability of the nomogram. The data points for error bar (m) and number of bootstrapping (B) are demonstrated in the figure legend (Figure 7).

## 4. Discussion

In multivariate Cox regression analysis, the RBC indices (mSLI, MCV, and MCH) were significant predictors of OS when evaluated individually in models incorporating age, ASA-PS, pleural invasion, TNM stage, and NUn score. Given its superior predictive performance, a model incorporating mSLI (mSLI model) was finalized. The mSLI model demonstrated improved predictive performance compared to both the baseline model and the intermediate model. Compared to the baseline model, the mSLI model showed significantly higher C-index values and superior discrimination at both 3 and 5 years, as reflected by higher AUCs. The IDI further supported its added prognostic value, and DCA demonstrated greater net clinical benefit in predicting OS. When compared to the intermediate model, the mSLI model exhibited slightly higher C-index and AUCs, though these differences were not statistically significant. However, the IDI still revealed meaningful improvements in discrimination, underscoring the model’s ability to enhance risk stratification.

Notably, mSLI remained independent of traditional risk factors and tumor-specific characteristics such as pleural/vascular/lymphatic/perineural invasion and TNM stage, as demonstrated by GAM analysis. This suggests that mSLI may capture distinct biological pathways or systemic factors that influence prognosis beyond those reflected by conventional clinical markers. Moreover, GAM analysis also demonstrated the independence of mSLI from established inflammatory and nutritional indices such as LMR, NLR, PLR, PNI, CAR, CLR, and CALLY scores, further supporting its unique prognostic value beyond commonly used composite markers [9,16,17,20,21,44]

The strong correlations between mSLI and its components, MCV and MCH, as demonstrated by Spearman’s correlation analysis, along with findings from the GAM analysis, underscore their substantial contributions to its predictive power and potential biological significance.

Elevated MCV is associated with increased all-cause mortality in the general population. A large cohort study of 36,260 individuals aged ≥40 years without cancer or anemia found that participants with higher MCV had an increased risk of all-cause mortality, particularly cancer [48]. In cancer-specific contexts, an elevated MCV has been consistently linked to poor prognosis. It is associated with a worse OS in patients with head and neck, esophageal, and gastroesophageal adenocarcinomas [49,50,51,52,53]. Additionally, an elevated MCV has been correlated with reduced disease-free survival in colorectal cancer (CRC) [54] and worse OS in lung cancer [32,33]. Similarly, MCH has been shown to independently predict OS in patients with gastroesophageal adenocarcinoma, esophageal cancer, and CRC [33,40,50,55], although the prognostic value of MCH in lung cancer is unclear. These findings suggest that the prognostic value of MCV and MCH may be attributable to underlying physiological or pathological conditions that influence both RBC indices and tumor behavior.

MCV and MCH levels are commonly linked to various conditions, including alcohol consumption, smoking, liver dysfunction, anemia, hypothyroidism, nutritional deficiencies, hematological malignancies, and the use of anticancer medications [56,57,58,59]. However, in the present study, GAM analysis indicated that alcohol use, smoking, liver dysfunction, anemia, and nutritional deficiencies—reflected by ALB levels and the PNI—were not significant predictors of the mSLI, suggesting that mSLI may be relatively independent of these traditional influencing factors. Although direct serum folate measurements were unavailable due to data limitations, previous studies involving 4448 patients have demonstrated a modest correlation between serum folate and ALB levels, the latter being a commonly used marker of nutritional status [60]. This raises the possibility that folate status may have been indirectly reflected through ALB levels. Nevertheless, direct assessment would be necessary to clarify its influence on mSLI. Additionally, due to the lack of available data, the potential influence of other factors such as vitamin B12 deficiency and thyroid dysfunction could not be evaluated, highlighting the need for further research to explore these associations.

MCV and MCH levels may reflect underlying alterations in the tumor microenvironment, particularly tumor-induced hypoxia, which is a well-established driver of cancer progression and treatment resistance in NSCLC [61,62]. Hypoxic conditions stimulate the production of erythropoietin, promoting erythropoiesis and accelerating the release of reticulocytes into circulation. Since reticulocytes are larger in size than mature RBCs, their increased presence can contribute to elevated MCV. These changes in RBC indices may therefore serve as indirect indicators of tumor-driven physiological stress and host response, providing additional insight into disease behavior and prognosis.

Tumor-driven cytokines, particularly interleukin-6 (IL-6), may disrupt normal erythropoiesis and contribute to alterations in MCV and MCH levels. IL-6 is known to influence RBC indices through its effects on iron metabolism, inflammation, and bone marrow function. However, the extent and nature of these effects may vary depending on underlying conditions and individual patient characteristics. Further research is warranted to clarify the specific mechanisms by which IL-6 and other cytokines influence erythropoiesis in the context of NSCLC and their role in shaping RBC-related prognostic markers such as mSLI [63,64,65].

In summary, the prognostic value of mSLI likely reflects a complex interplay of systemic physiological conditions, tumor-induced alterations in erythropoiesis, and inflammatory signaling pathways. Alternatively, it is also plausible that MCV and MCH contribute directly to cancer development and progression through as-yet undefined biological mechanisms. Continued investigation into these underlying pathways may not only deepen the biological understanding of mSLI but also enhance its clinical applicability as a prognostic tool in patients with NSCLC.

In addition to mSLI, other significant predictors of OS included age, ASA-PS, pleural invasion, TNM stage, and NUn score. Age, a well-established prognostic factor, is associated with poorer outcomes due to comorbidities and diminished physiological reserve [3,5,10,66]. Higher ASA-PS scores, reflecting worse preoperative physical status, were correlated with increased perioperative risk and poorer survival [66]. TNM staging continues to serve as the fundamental framework for NSCLC prognostication, with higher stages associated with diminished survival owing to the reduced feasibility of complete surgical resection [3,5,6,9,10]. Pleural invasion is another strong predictor of more advanced disease and higher recurrence risks [3,10]. The NUn score, which integrates CRP, ALB, and WBC values, was initially developed to predict anastomotic leakage in esophageal cancer [44]. However, our previous study highlighted its broader prognostic utility in NSCLC, as it independently predicts OS [19].

While indices such as LMR, NLR, PNI, CAR, CLR, and CALLY demonstrated statistical significance in univariate analyses, they did not retain significance in multivariate models, indicating limited independent prognostic relevance. These findings differ from those of previous studies, potentially due to variations in cohort characteristics and analytical methods, particularly the use of continuous variables without categorization [9,16,17,20,21,44]. Dichotomizing continuous variables, which is common in clinical research, often results in a significant loss of information and potential misrepresentation of variable-outcome relationships. Moreover, the use of an “optimal” cutpoint—typically chosen to minimize the *p* value—increases the risk of spurious significance, overestimates differences between groups, and produces inaccurately narrow confidence intervals, further undermining the reliability of such approaches [45,46,47].

The mSLI model, encompassing significant variables like mSLI, age, ASA-PS, pleural invasion, TNM stage, and NUn score, offers meaningful clinical applicability for guiding treatment decisions in stage I–IIIA NSCLC. Low VIF values confirmed the minimal collinearity among these variables, thus ensuring the robustness of the model. Fractional polynomial analysis revealed linear relationships between continuous variables (age, mSLI, and NUn scores) and log-relative hazards, indicating proportional effects on mortality risk.

The mSLI model demonstrated significantly enhanced predictive performance compared to the baseline TNM staging model, which is the current standard for assessing NSCLC prognosis, as evidenced by improvements across multiple metrics, including the C-index, AUC values, IDI, and DCA. These findings highlight the added value of incorporating host-related factors—such as systemic inflammation and nutritional status—into survival prediction, which are not captured by anatomical staging alone. While TNM staging remains a cornerstone of cancer classification, the mSLI model offers meaningful complementary prognostic information that can support more nuanced risk stratification. Although the improvements in C-index and AUC were statistically modest, their consistency across multiple metrics and time points reinforces the potential clinical utility of mSLI, particularly in guiding long-term follow-up and personalized management.

The mSLI model demonstrated slightly improved predictive performance compared to the intermediate model. Although the differences in C-index and AUC values did not reach statistical significance, the consistently higher values observed with the mSLI model indicate a trend toward enhanced discrimination. Importantly, the IDI metric provided compelling evidence for the added prognostic value of mSLI, showing statistically significant gains in risk discrimination. These results underscore the model’s ability to refine risk stratification and enhance individualized patient assessment. The improved IDI suggests that mSLI may offer meaningful clinical utility, particularly in identifying patients with distinct risk profiles who may benefit from tailored follow-up or treatment strategies. While the numerical differences in traditional metrics were modest, the consistent direction of improvement supports the value of incorporating mSLI into clinical decision-making frameworks.

Taken together, these findings suggest that the mSLI model is a practical and informative complement to TNM staging, offering more nuanced and individualized survival prediction. Its simplicity and accessibility further support its integration into routine clinical practice. Further external validation will be important to confirm its broader applicability and to solidify its role alongside existing prognostic tools.

The accompanying nomogram based on the mSLI model enables personalized risk assessment and identifies high-risk patients who may benefit from more aggressive treatment, closer monitoring, or adjuvant therapies, even in early-stage disease. Conversely, low-risk patients can avoid unnecessary interventions, reduce treatment-related side effects, and improve their quality of life. Furthermore, the model aids in optimizing follow-up strategies by tailoring the surveillance intensity based on individualized risk levels. By offering practical three- and five-year survival predictions, the nomogram serves as a valuable tool for patient counseling and treatment planning, reinforcing its potential for clinical integration.

A major strength of this study lies in its novelty, being the first to evaluate the prognostic value of the mSLI in patients with stage I–IIIA NSCLC undergoing curative-intent resection. This offers important new insights into its potential clinical utility. Another strength is the use of a comprehensive dataset encompassing a broad spectrum of clinicopathological and laboratory variables, which enhances the robustness and credibility of the findings. Rigorous statistical methodologies, including bootstrap resampling with 1000 iterations, were employed to ensure the reliability and generalizability of the results. Notably, mSLI was treated as a continuous variable rather than dichotomized, preserving the full range of data, reducing the risk of overfitting—particularly given the moderate sample size—and enabling more individualized prognostic assessments. This approach also improved the model’s generalizability and statistical power. Fractional polynomial analysis confirmed linear relationships between continuous variables (age, mSLI, and NUn score) and log-relative hazards, supporting the proportional hazards assumption. Furthermore, a nomogram based on the mSLI model was developed, offering a practical and clinically relevant tool for predicting OS and guiding personalized patient care. Importantly, the key diagnostic parameters used in this model—MCV and MCH—are part of standard preoperative laboratory assessments, reinforcing the model’s feasibility and cost-effectiveness in routine clinical practice.

However, this study has some limitations. Because this was a retrospective analysis, it was inherently vulnerable to bias, which may have affected the generalizability of the findings. Being conducted at a single institution, the results may not be fully representative of broader, more diverse populations. Notably, the study cohort was predominantly composed of Asian patients, with only a small proportion of Caucasian patients, which may limit the applicability of the mSLI model to multiethnic NSCLC populations. Given that racial and ethnic differences can influence disease characteristics and prognosis, further validation using external, ethnically diverse, and globally representative datasets is essential. Although we employed internal validation using robust statistical techniques, including bootstrap resampling, the absence of external validation using independent cohorts weakened the applicability of the model across diverse populations. Additionally, unmeasured confounding variables may have influenced the outcomes, potentially affecting the predictive accuracy of the model.

## 5. Conclusions

The mSLI emerged as an independent predictor of survival in multivariate Cox regression analysis. The mSLI model, which incorporated mSLI alongside age, ASA-PS, pleural invasion, TNM stage, and the NUn score, demonstrated significantly improved prognostic performance compared to the standard TNM staging system, with consistent gains across multiple metrics, including C-index, AUC, IDI, and DCA. While its advantage over the intermediate model was modest, the consistently higher C-index and AUC values suggest a trend toward better risk discrimination. Notably, the IDI provided robust evidence of mSLI’s added prognostic value.

Nonetheless, several limitations must be acknowledged. The retrospective nature of the study, its single-institution design, and a predominantly Asian patient population may limit the generalizability of these findings. Additionally, the absence of external validation restricts the applicability of the model across broader and more diverse NSCLC populations.

Future studies should aim to clarify the biological mechanisms underlying mSLI to enhance its clinical relevance. Validation in larger, multicenter, and multiethnic cohorts is necessary to assess generalizability. Incorporating mSLI with emerging molecular and immune biomarkers may further refine prognostic accuracy, while evaluating its impact on clinical decision-making—such as tailoring adjuvant therapy or surveillance—will help establish its real-world utility.

In conclusion, the mSLI model offers a promising enhancement to NSCLC prognostication by improving risk stratification beyond traditional models. With further validation, it holds potential to support more personalized treatment planning and long-term patient management.

## Figures and Tables

**Figure 1 biomedicines-13-00937-f001:**
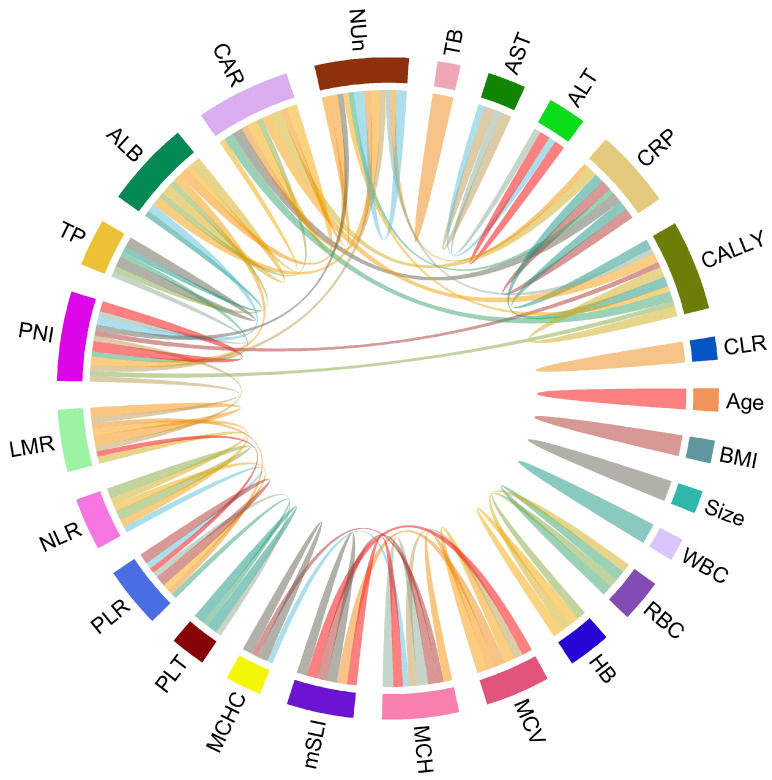
Chord diagram of correlations among variables. The chord diagram visually represents the correlations with either >0.5 or <−0.5 between variable pairs, indicating significant associations.

**Figure 2 biomedicines-13-00937-f002:**
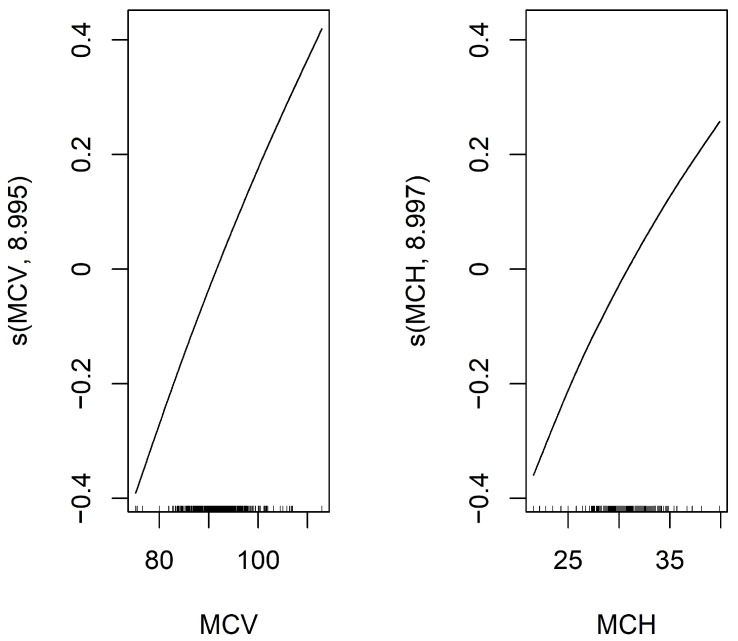
Smooth functions of MCV and MCH on mSLI in the Generalized Additive Model (GAM). This figure displays the smooth functions of MCV and MCH from a GAM with a smoothing spline using 9 degrees of freedom. The Y-axis shows the estimated effect of MCV and MCH on the response variable, mSLI, after adjusting for other covariates. The black line represents the fitted smooth function, illustrating how MCV and MCH influence mSLI.

**Figure 3 biomedicines-13-00937-f003:**
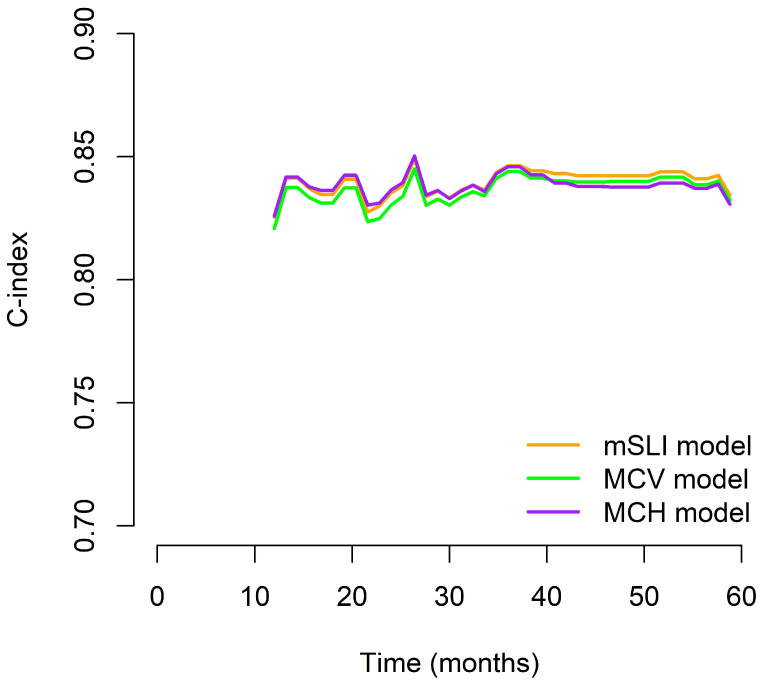
Time-dependent C-index curves comparing the prognostic performance of the mSLI, MCV, and MCH models over five years.

**Figure 4 biomedicines-13-00937-f004:**
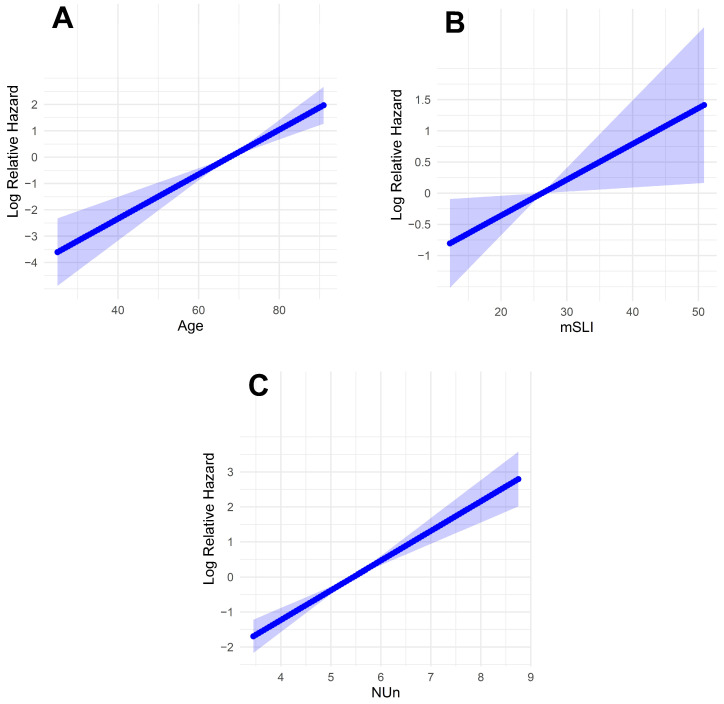
Log-relative hazard models. (**A**) Model for age; (**B**) model for modified Shine and Lal index; (**C**) model for Noble and Underwood Score. The shaded areas indicate 95% confidence intervals.

**Figure 5 biomedicines-13-00937-f005:**
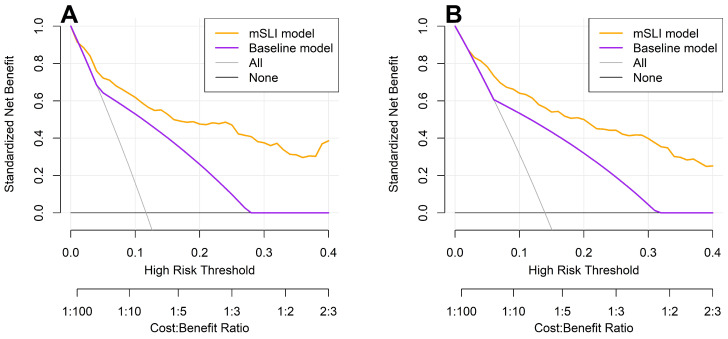
Decision curve analysis for 3-year (**A**) and 5-year (**B**) overall survival: a comparison between the mSLI and baseline models. The mSLI model incorporates age, ASA-PS, pleural invasion, TNM stage, mSLI, and NUn score. The baseline model incorporates only the TNM staging system.

**Figure 6 biomedicines-13-00937-f006:**
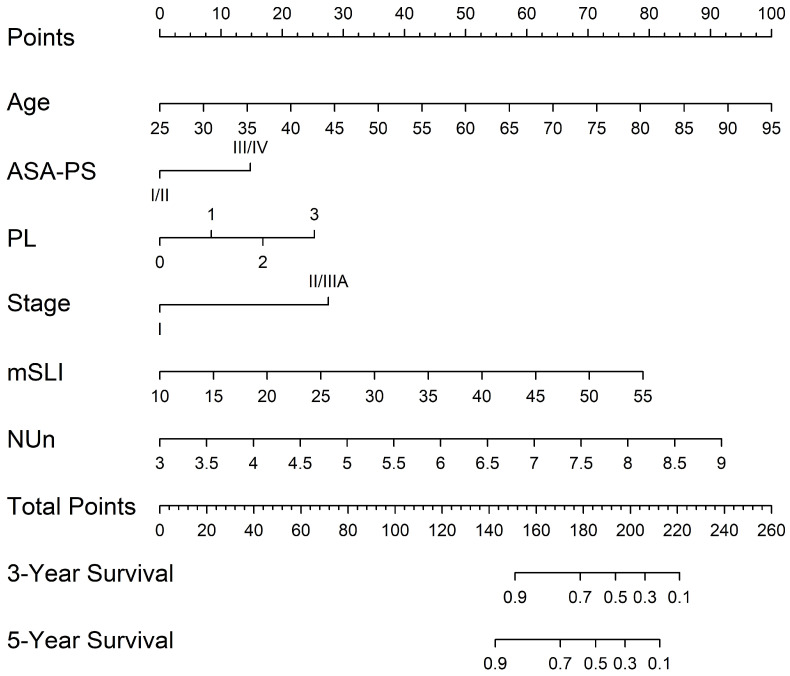
Nomogram constructed from the mSLI model for predicting 3- and 5-year overall survival. The nomogram illustrates the predicted probability of overall survival using the mSLI model, which incorporates age, ASA-PS, pleural invasion (PL), TNM stage, mSLI, and NUn score.

**Figure 7 biomedicines-13-00937-f007:**
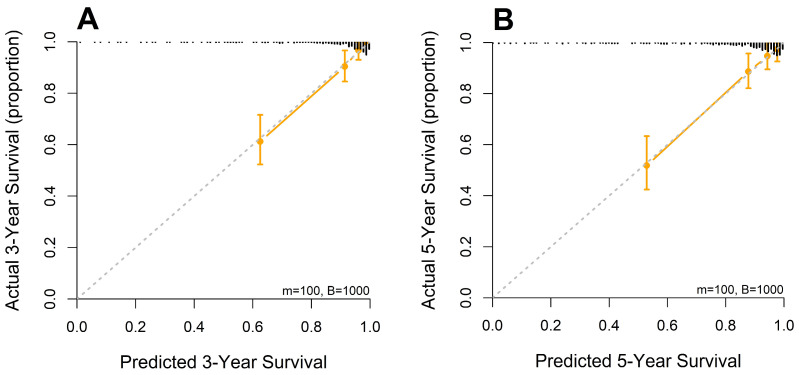
Calibration curves for 3-year (**A**) and 5-year (**B**) overall survival predicted by the mSLI model. The mSLI model incorporates age, ASA-PS, pleural invasion, TNM stage, mSLI, and NUn score. The data points for error bar (m) and number of bootstrapping (**B**) are demonstrated in the figure legend.

**Table 1 biomedicines-13-00937-t001:** Baseline characteristics of patients with non-small cell lung cancer.

Variables	*n*	% or Median (IQR)	Variables	*n*	% or Median (IQR)
Age, years	437	69.0 (62.0–74.0)	Perineural invasion		
Sex			No	430	98.4%
Men	257	58.8%	Yes	7	1.6%
Women	180	41.2%	TNM Stage		
Smoker			IA/IB	300	68.6%
Never	247	56.5%	IIA/IIB	79	18.1%
Current/Past	190	43.5%	IIIA	58	13.3%
Alcohol consumption			WBC, per μL	437	6290 (5280–7400)
No	325	74.4%	RBC, ×10^6^ per μL	437	4.3 (3.9–4.6)
Yes	112	25.6%	Hemoglobin, g/dL	437	13.2 (12.1–14.1)
ASA-PS			mSLI	437	25.8 (23.9–28.4)
I/II	360	82.4%	MCV, fL	437	91.4 (88.8–94.4)
III/IV	77	17.6%	MCH, pg	437	30.8 (29.9–32.1)
BMI, kg/m^2^	437	23.8 (21.8–26.0)	MCHC, g/dL	437	33.7 (33.1–34.4)
Histology			Platelet, ×10^3^ per μL	437	236 (200–277)
Squamous	95	21.7%	LMR	437	3.8 (2.9–4.9)
Non-squamous	342	78.3%	NLR	437	2.0 (1.5–2.8)
Type of surgery			PLR	437	130.3 (102.4–161.2)
Segmentectomy	143	32.7%	Total protein, g/dL	437	7.2 (6.8–7.5)
Lobectomy	288	65.9%	Albumin, g/dL	437	4.2 (3.9–4.4)
Pneumonectomy	6	1.4%	Bilirubin, mg/dL	437	0.5 (0.4–0.7)
Size, cm	437	2.5 (1.7–3.5)	AST, U/L	437	22.0 (19.0–27.0)
Pleural invasion			ALT, U/L	437	16.0 (12.0–23.0)
0	340	77.8%	CRP, mg/dL	437	0.1 (0.1–0.3)
≥1	97	22.2%	CAR	437	0.03 (0.02–0.08)
Vascular invasion			PNI	437	51.2 (47.7–53.7)
No	413	94.5%	CLR	437	0.6 (0.4–1.8)
Yes	24	5.5%	CALLY index	437	6.7 (2.3–11.2)
Lymphatic invasion			NUn score	437	5.4 (5.0–5.8)
No	381	87.2%			
Yes	56	12.8%			

**Table 2 biomedicines-13-00937-t002:** Univariate Cox regression analysis for overall survival.

Variables *	HR (95% CI)	*p* Value
Age, years	1.09 (1.06–1.12)	<0.001
Sex (women vs. men)	0.39 (0.23–0.68)	<0.001
Smoker (current/former vs. never)	2.59 (1.58–4.23)	<0.001
Alcohol consumption (present vs. absent)	0.88 (0.52–1.50)	0.644
ASA-PS (III/IV vs. I/II)	3.11 (1.93–5.01)	<0.001
BMI, kg/m^2^	0.99 (0.92–1.06)	0.823
Histology (squamous vs. non-squamous)	3.87 (2.45–6.11)	<0.001
Tumor size, cm	1.31 (1.19–1.44)	<0.001
Pleural invasion **	1.95 (1.54–2.46)	<0.001
Vascular invasion (present vs. absent)	2.43 (1.21–4.89)	0.013
Lymphatic invasion (present vs. absent)	2.00 (1.15–3.49)	0.014
Perineural invasion (present vs. absent)	3.54 (1.11–11.29)	0.033
TNM stage (II/IIIA vs. I)	5.11 (3.14–8.33)	<0.001
WBC, per μL	1.00 (1.00–1.00)	0.001
Anemia (present vs. absent)	1.33 (0.83–2.12)	0.233
Modified Shine and Lal Index	1.06 (1.01–1.12)	0.027
MCV, fL	1.05 (1.00–1.10)	0.068
MCH, pg	1.10 (0.98–1.24)	0.111
MCHC, g/dL	1.02 (0.81–1.29)	0.840
Platelet, ×10^3^ per μL	1.00 (1.00–1.00)	0.860
LMR	0.75 (0.64–0.89)	<0.001
NLR	1.34 (1.17–1.55)	<0.001
PLR	1.00 (1.00–1.01)	0.170
Albumin, g/dL	0.15 (0.08–0.29)	<0.001
CRP, mg/dL	1.17 (1.11–1.22)	<0.001
CAR	1.73 (1.45–2.07)	<0.001
PNI	0.88 (0.84–0.92)	<0.001
CLR	1.02 (1.01–1.03)	<0.001
CALLY	0.90 (0.86–0.94)	<0.001
NUn score	2.33 (1.84–2.96)	<0.001

* Reference values are shown in parentheses on the right. ** Ordinal variables.

**Table 3 biomedicines-13-00937-t003:** Comparison of hazard between mSLI, MCV and MCH models.

Variables *	mSLI Model (HR, 95% CI)	*p* Value	MCV Model (HR, 95% CI)	*p* Value	MCH Model (HR, 95% CI)	*p* Value
Age, years	1.07 (1.03–1.10)	<0.001	1.06 (1.03–1.10)	<0.001	1.06 (1.03–1.10)	<0.001
ASA-PS (III/IV vs. I/II)	1.92 (1.18–3.15)	0.009	2.01 (1.23–3.28)	0.005	1.85 (1.12–3.04)	0.016
Pleural invasion **	1.45 (1.14–1.84)	0.002	1.44 (1.14–1.83)	0.002	1.45 (1.14–1.84)	0.002
TNM stage (II/IIIA vs. I)	3.37 (2.01–5.66)	<0.001	3.39 (2.02–5.69)	<0.001	3.38 (2.01–5.68)	<0.001
NUn score	1.96 (1.50–2.58)	<0.001	1.92 (1.47–2.52)	<0.001	1.99 (1.51–2.62)	<0.001
mSLI	1.08 (1.03–1.14)	0.003	-	-	-	-
MCV, fL	-	-	1.06 (1.01–1.12)	0.012	-	-
MCH, pg	-	-	-	-	1.18 (1.05–1.34)	0.007

* Reference values are shown in parentheses on the right. ** Ordinal variables. The mSLI model demonstrates a slightly higher C-index compared to the MCV model (0.840 vs. 0.837, *p* = 0.430) and the MCH model (0.840 vs. 0.837, *p* = 0.828); however, these differences are not statistically significant.

**Table 4 biomedicines-13-00937-t004:** Comparative evaluation of predictive performance for overall survival using the mSLI, intermediate, and baseline models across multiple metrics.

Metrics	mSLI Model (MM)	Intermediate Model (IM)	Baseline Model (BM)	MM vs. BM (Difference)	*p* Value	MM vs. IM (Difference)	*p* Value
C-index	0.840 (0.023)	0.835 (0.021)	0.708 (0.028)	0.135 (0.021)	<0.001	0.006 (0.009)	0.554
AUC 3Y	0.878 (0.027)	0.873 (0.026)	0.734 (0.034)	0.143 (0.024)	<0.001	0.005 (0.010)	0.614
AUC 5Y	0.850 (0.033)	0.846 (0.032)	0.708 (0.036)	0.143 (0.029)	<0.001	0.004 (0.010)	0.662
IDI 3Y				0.241 (0.043)	<0.001	0.047 (0.026)	0.008
IDI 5Y				0.218 (0.043)	<0.001	0.029 (0.021)	0.020

Values shown in parentheses correspond to standard errors. The mSLI model incorporated age, ASA-PS score, pleural invasion, TNM stage, mSLI, and the NUn score. The intermediate model integrated age, ASA-PS score, pleural invasion, TNM stage, and the NUn score. The baseline model incorporates only the TNM staging system.

## Data Availability

The datasets presented in this study are available upon request from the corresponding author due to ethical reasons.

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
