# Peer review of "Prognostic Significance of Modified Shine and Lal Index in Patients with Non-Small Cell Lung Cancer Undergoing Surgical Resection"

_biomedicines, 2025, doi:10.3390/biomedicines13040937_

Round 1
Reviewer 1 Report
Comments and Suggestions for Authors
This work defines the use of multivariate Cox regression analysis for mean corpuscular volume (MCV), mean corpuscular hemoglobin (MCH), and modified Shine and Lal Index (mSLI). It can be published after the queries have been addressed.
- The introduction must be re-framed by justifying the need for this research, why 437 patients were chosen, and why Cox regression analysis.
- Provide the abbreviations for all terminologies while presenting first. This could improve the readability. Though the author delivers the abbreviations at the end, this is essential.
- Where is the age in Table 1?
- In-depth critical discussion for Tables 1 - 4 with merits and limitations must be added. Comparative accounts for sections 3.4 and 3.5 must be improved with added discussion.
- The conclusion section lacks merits, limitations, and future directions.
- Improve the resolution of all the figures and mention the number of data points for the error bar containing figures.
- Update the recent and relevant literature
Moderate English correction and spell check is required
Reviewer 2 Report
Comments and Suggestions for Authors
This study addresses an important gap by investigating the prognostic role of understudied RBC indices (MCV, MCH, MCHC) and the derived mSLI in early-stage NSCLC. The focus on readily available, low-cost hematologic markers adds translational value to risk stratification. The use of multivariable Cox regression, C-index comparisons, and IDI analysis strengthens the statistical validity of the findings. The mSLI model’s superiority over TNM staging alone (C-index: 0.840 vs. 0.708) is compelling. The proposed mSLI integration could refine postoperative prognostic frameworks, aiding in personalized follow-up or adjuvant therapy decisions. But I have several following concerns:
- While the statistical association is clear, the biological rationale linking mSLI (reflecting RBC properties) to NSCLC outcomes remains speculative. A brief discussion on potential mechanisms (e.g., chronic hypoxia, systemic inflammation, or iron metabolism) would strengthen the manuscript.
- The retrospective, single-center design limits generalizability. As noted by the authors, external validation in independent cohorts is essential before clinical adoption.
- Were RBC-altering conditions (e.g., anemia, nutritional deficiencies) accounted for? Clarifying how these were addressed (e.g., exclusion criteria or adjustments) would mitigate potential bias.
- While IDI improvements are statistically significant, their absolute magnitude (e.g., 3-/5-year IDI values) should be contextualized for clinicians to assess real-world impact.
- Please use the standard three-line form.
- Please unify the format of references in the article, including the author's name, the case of words in the title of the article, the writing of the name of the journal, and the page number.
The English could be improved to more clearly express the research.
Reviewer 3 Report
Comments and Suggestions for Authors
In this manuscript, the authors revealed that mSLI is an independent marker for OS in NSCLC, providing incremental predictive value beyond that of traditional models. In the test, mSLI outperformed other models and demonstrated significant improvements in IDI at 3 and 5 years. Meanwhile, the mSLI model achieved a higher C-index than the intermediate model (which included age, ASA-PS, pleural invasion, TNM stage, and NUn score).
In summary, this research provides valuable insights into the mSLI model and its clinical applications.
Regarding the mSLI model, are there any independent datasets available to validate its advantages? If so, please discuss them in the discussion section.
People of different races are important to the disease prediction. To enhance the model's applicability, it is essential to consider the dataset’s generalizability. Is the model applicable to a global NSCLC dataset? If so, could the authors add a related discussion in the discussion section?
Reviewer 4 Report
Comments and Suggestions for Authors
The article titled “Prognostic Significance of the Modified Shine and Lal Index in Patients with Non-Small Cell Lung Cancer Undergoing Surgical Resection” aimed to study the prognostic significance of RBC indices in NSCLC patients with surgical resection. The article is informative, and the methodology is appropriate. However, the presentation of the results in the paper is not very clear and looks a bit confusing. Here are some detailed comments:
- In line 250-251, how was the C-index of the mSLI generated? Why wasn’t this value showed in table 3? The authors should explain the difference and research significance between mSLI as a parameter and MCV and MCH as independent parameters.
- It is recommended that the author make a regression curve to predict the correlation between mSLI, MCV, and MCH values ​​and patient prognosis and survival rate, which will make the results more direct.
- What is the research significance of the vertical axis in Figure 3?
- In Figure 4, although mSLI has a higher net benefit than other models, how to derive mSLI to better predict 3- and 5-year OS. What is the relationship between net benefit and OS.
- In line 475, “6. The Patents” subheading looks like an error. Please check.
Round 2
Reviewer 1 Report
Comments and Suggestions for Authors
I appreciate the author's effort in improving this manuscript.
Reviewer 2 Report
Comments and Suggestions for Authors
The authors have addressed all my comments, I recommend accepting it in current form.
Reviewer 4 Report
Comments and Suggestions for Authors
This manuscript has been sufficiently improved to warrant publication in Biomedicines